# Breaking Dogmas in Axillary Lymphadenectomy and Quality of Life

**DOI:** 10.3390/cancers17132201

**Published:** 2025-06-30

**Authors:** Sandra López Gordo, Jaime Jimeno-Fraile, Anna García-Monferrer, Pau Nicolau, Neus Ruiz-Edo, Elena Ramirez-Maldonado, Santiago Rojas, Cristina Serra-Serra

**Affiliations:** 1General and Digestive Surgery Department, Maresme Health Consortium (Mataró Hospital), 08304 Mataró, Spain; agarciamo@csdm.cat (A.G.-M.); mruized@csdm.cat (N.R.-E.); 2Unit of Human Anatomy and Embriology, Department of Morphological Sciences, Faculty of Medicine, Universitat Autònoma de Barcelona, 08193 Cerdanyola del Vallès, Spain; santiago.rojas@uab.cat; 3Breast Unit, General and Digestive Surgery, Marqués de Valdecilla, 39008 Santander, Spain; jaime.jimeno@scsalut.es; 4Department of Medical and Surgical Sciences, University of Cantabria, Marqués de Valdecilla, 39008 Santander, Spain; 5Ginecology Department, Maresme Health Consortium (Mataró Hospital), 08304 Mataró, Spain; pnicolau@csdm.cat (P.N.); cserra@csdm.cat (C.S.-S.); 6General and Digestive Surgery Department, Universitary Hospital of Tarragona Joan XXIII, 43005 Tarragona, Spain; rramirez.hj23.ics@gencat.cat; 7Biomedicine Department, Rovira i Virgili University, 43002 Tarragona, Spain

**Keywords:** breast cancer, axillary lymphadenectomy, drain, sealant, ambulatory, minimal invasive surgery, quality of life

## Abstract

Axillary lymphadenectomy for breast cancer is becoming less necessary; however, for now, it must be performed in certain situations, such as a high axillary burden. This review aims to examine the new modifications and innovations compared with the classic technique and to explain why some dogmas still persist today that are difficult to overcome. Outpatient surgery, the placement of sealants and drains, and minimally invasive surgery are discussed, along with a brief review of the quality of life of these patients. We will review the previously mentioned topics, attempting to address all the historical aspects and results of the techniques that have always been used, and why they have seemed to resist modification over the years.

## 1. Introduction

Axillary lymph node dissection (ALND) in breast cancer (BC) is becoming less common; however, it is still necessary in certain cases, like a high axillary volume. The axillary tumor burden is generally assessed through a combination of a clinical examination and imaging modalities such as ultrasound and MRI, and a histopathological confirmation is always required before the indication of performing an axillary lymphadenectomy. When performed, a traditional approach is often followed in the surgical technique and postoperative care. Despite evolving practices, certain aspects of this procedure continue to be handled in a conventional manner in most hospitals.

One of the most prevalent postoperative complications following ALND is seroma formation, reported in 15–90% of cases, with 0–26% of presenting as a symptomatic seroma after lymphadenectomy or sentinel lymph node dissection [1,2,3,4]. Seroma results from an accumulation of serous fluid in the dead space created by tissue removal, leading to patient discomfort, an increased risk of infection, and potential delays in adjuvant therapy. The seroma volume is one of the main factors that causes discomfort in patients undergoing axillary surgery for BC. However, many studies describe the total seroma volume by measuring the accumulated milliliters, regardless of symptoms. However, the volume itself should be considered less relevant than the symptoms and expressed by patient discomfort. Patients’ symptoms should be the signal to indicate further clinical intervention.

With the aim to reduce seroma formation, surgical drains have traditionally been used to evacuate fluid and promote tissue adherence. However, in recent years, new approaches have challenged this standard practice, including early drain removal, no-drain techniques, and the use of fibrin sealants or axillary padding.

Other critical aspects are the possibility of same-day discharge, which has been widely integrated into many surgical specialties but is less common after ALND. Similarly, minimal invasive surgery, although adopted in many fields years ago, has not yet been fully consolidated in BC surgery, despite being described for some time.

In this context, quality of life (QoL) is a basic outcome to consider after BC surgery and especially after ALDN. ALDN impacts in arm function, pain, and lymphedema, factors closely related to QoL. The instruments used to evaluate QoL after BC are EORTC QLQ-C30, QLQ-BR23, FACT-B, and SF-36; these instruments asses physical, emotional and psychosocial domains. Interestingly, while ALND clearly increases arm-related symptoms compared to sentinel node biopsy, its overall impact on global QoL scores appears limited, likely due to the overlapping effects of systemic therapies and breast surgery. As surgical techniques evolve, improving QoL must remain a central objective in the management of BC patients undergoing axillary surgery.

This review sought to discuss the impact on QoL after ALDN, the use of sealants or drains, same-day discharge, and minimal invasive surgery for BC.

## 2. Materials and Methods

We conducted a comprehensive literature search using predefined keywords across several databases to identify relevant publications, covering publications from January 1980 to December 2024. This is not a systematic review; no formal risk of bias assessment or PRISMA flow diagram was applied. We identified studies that included the terms “axillary lymphadenectomy” OR “axillary dissection” ALND (“breast cancer” OR “surgical management” OR “oncological outcomes” OR “ambulatory surgery” OR “no drain” OR “sealant-free”).

To ensure a broad and balanced perspective, we included peer-reviewed articles, systematic reviews, meta-analyses, and clinical guidelines published in English. Additionally, we employed a snowballing technique, reviewing the reference lists of selected articles to identify further relevant studies. Each article was independently assessed by a member of the research team for relevance, with a focus on studies discussing the evolution of axillary surgery, de-escalation strategies, ambulatory lymphadenectomy, the absence of drains or sealants, and their impacts on oncological safety and patient outcomes. Special attention was given to studies evaluating postoperative recovery, complications, and QoL following ALDN.

## 3. Ambulatory Surgery and Early Discharge After ALND

Traditionally, all patients undergoing BC surgery required hospital admission. However, same-day discharge has progressively become more common, especially in procedures involving the breast such as lumpectomies and mastectomies. Nevertheless, some surgeries still have high hospitalization rates, such as immediate breast reconstructions and ALDN. Following this trend, a patient undergoing a mastectomy with ALDN is most likely to suffer an extended hospitalization. There is evidence in literature that same-day surgery after mastectomy and axillary clearance is feasible and does not increase postoperative complications. The authors emphasized that proper patient selection and education significantly reduce hospital stays and associated costs [5].

One of the main barriers to ambulatory discharge in BC surgery is patient belief, with studies reporting up to 20% of patients refusing early discharge [6]. This resistance could appear because of the lack of information given by the medical team, feelings of insecurity toward the diagnosis or the procedure, or a fear of managing postoperative care at home.The role of case management nurses and the surgeons’ dedication to explaining outpatient discharge pathways, preoperative education, and shared decision-making approaches can improve patient confidence and thus facilitate the acceptance of early discharge.

Major complications within the first few hours post-surgery are luckily uncommon, with bleeding being the most frequent serious complication requiring reintervention [7]. A fear of such complications might lead some patients to request hospital admission instead of opting for ambulatory management.

Other common reasons for prolonged hospitalization include postoperative nausea and pain, although pain management has significantly improved with the use of regional anesthesia techniques such as the Breast Interfascial Regional Block (BRILMA) or Pectoral Blocks (PECS) [8]. Effective multimodal pain control is instrumental for enabling same-day discharge in this scenario.

For a secure and efficient early discharge, certain criteria must be met, including adequate home support (a caregiver for the first night), proximity to a hospital (living within 100 km or 1 h from a medical facility), or access to a mobile phone for emergency communication [6]. These measures are essential to ensure the early detection and prompt management of potential complications.

### 3.1. Benefits of Ambulatory Surgery

Level I and II axillary dissection can be safely performed as an outpatient procedure with [9] or without drains [10]. Ambulatory surgery offers numerous advantages, such as increased patient autonomy and QoL, lower healthcare costs [11], improved physical recovery, including better shoulder mobility and reduced postoperative pain at three months [12] and potentially lower seroma rates (18% vs. 34%, *p* < 0.001), without increasing other complications [13].

### 3.2. The Role of ERAS Protocols in Facilitating Early Discharge

Enhanced Recovery After Surgery (ERAS) protocols have played a pivotal role in enabling safe and effective ambulatory surgery for breast cancer (BC) patients. These protocols include evidence-based perioperative strategies such as minimized preoperative fasting, multimodal pain management with opioid reduction, optimized fluid management, and early mobilization [8,14]. Regional anesthesia techniques, such as PECS blocks and serratus plane blocks, combined with light general anesthesia are also included and help with further enhanced recovery.

### 3.3. Early Discharge Feasibility and Alternatives to Drains

Most studies evaluating ambulatory ALDN have involved patients discharged with a drain [15,16]. While drains are generally safe and effective, they can cause discomfort and challenges in daily activities and self-care.

A randomized clinical study by Horgan et al. compared early discharge on postoperative day 3 with a drain in situ versus a standard 7-day hospitalization. Their findings revealed no significant differences in seroma formation, wound infection, or psychological outcomes. Moreover, patient satisfaction was high in the early discharge group, and this approach led to substantial healthcare cost savings [16].

Given that symptomatic seroma is a common side effect of ALDN, some researchers have explored drain-free surgical techniques. The “padding” technique, which reduces the axillary dead space without using drains, has shown promising results [6,17]. Classe et al. further evaluated this technique in a randomized trial that included an ambulatory group, achieving an ambulatory surgery rate of 84.5% with symptomatic seroma rates of 22.2%, comparable to previous studies (18%) [6].

Today, less invasive surgical techniques continue to emerge, aiming to further minimize symptomatic seroma formation and improve recovery; this aspect will be discussed in later sections of the present review.

### 3.4. Economic Impact of Ambulatory Surgery

The economic impact of ambulatory surgery in BC treatment is significant, as reducing hospital stays leads to substantial cost savings without compromising patient outcomes. In the United Kingdom, a study by Holcombe et al. estimated an annual savings of £187,500, based on the assumption that a hospital bed costs GBP 250 per night and that approximately 75% of patients could benefit from early discharge [13]. Extending this analysis, a broader economic evaluation suggested that reducing hospital stays by 5–6 days could lower costs by approximately GBP 1000 per patient, generating substantial savings for the National Health Service [12]. Similarly, in Europe, a study conducted in the Netherlands identified postoperative hospitalization as the most significant expense after surgical BC treatment. By implementing early discharge protocols, hospitals were able to reduce costs by EUR 1320 per patient, this facilitates a reduction of a 30% in total postoperative care costs [11]. These findings highlight the financial viability of shifting toward ambulatory care models, reinforcing the need for the further adoption of early discharge strategies in BC surgery.

## 4. Omission of Surgical Drains in Lymphadenectomy

The use of drains in ALDN for breast cancer remains a widely debated topic in surgical practice. The primary goals of drains are to control postoperative seroma formation and avoid punctures in symptomatic seromas. Seroma formation rates vary between 3% and 85% across studies, with prolonged drainage potentially delaying adjuvant therapies and increasing infection risks [18].

### 4.1. Types of Drainage Protocols

The most commonly used drains in ALND are closed suction drains, which function by continuously evacuating fluid from the surgical site through a vacuum-assisted mechanism. Standard drain care includes securing the drain to prevent dislodgement, daily monitoring of fluid output, and vigilance for signs of infection or blockage.

Volume-controlled drainage protocols, where the drain is removed once the daily output falls below 30–50 mL, has been the standard practice [19]; this may take up to 5 to 15 days. Comparatively, short-term drainage protocols advocate for removal within 24 to 48 h postoperatively, regardless of output. One of the main issues with prolonged drainage placement is inflammation. In the presence of an inflammatory exudate, the drain itself may exacerbate the inflammatory response, potentially increasing seroma formation rather than preventing it [2]. In addition, prolonged drainage can cause patient discomfort and increase the risk of an infection.

### 4.2. Use of Drains

The omission of drains is not a new trend; in fact, it has been practiced since 1988. Cameron et al. (1988) [20], Somers (1992) et al. [21], Zavotsky et al. (1998) [22], and Jain et al. (2004) [2] compared volume-controlled drainage with no drainage and reported a higher incidence of clinically significant seroma formation in patients who did not receive drains; this fact seems reasonable and logical, given that drainage promotes the elimination of the formed seroma, preventing its accumulation. While larger seroma volumes may prolong recovery and cause discomfort, the current literature does not strongly support a link between the seroma volume and long-term oncological outcomes.

Authors, in 1990, reported the results of a study of 259 consecutive patients who underwent ALDN without drainage or sealants, performed as an outpatient procedure. In their series, symptomatic seroma occurred in only 4.2% of cases (Levels I and II). These findings, dating back decades, demonstrate that symptomatic seroma without drainage has a low incidence and is entirely manageable [10].

Despite the increased seroma incidence, these studies found no significant differences in wound infection rates between groups, concluding that drains may reduce seroma occurrence but they do not necessarily have an impact on the infection risk. Additionally, avoiding drains was associated with a shorter hospital stay [23,24], improved patient comfort, and more efficient healthcare resource utilization.

Avoiding the placement of a drain has also shown to improve pain control [22]; therefore, this could help in achieving a better QoL and higher autonomy after ALDN.

Studies have shown that closed suction drainage can help prevent excessive seroma accumulation. A meta-analysis comparing volume-controlled drainage (<30–50 mL/24 h) to no or short-term drainage found that seroma formation was more common in patients without drains (RR 0.44, 95% CI 0.24–0.80). However, while drains reduced the need for repeated aspirations, their presence was associated with longer hospital stays [19] and less infection [3,25]. A systematic review published in 2010 also found a reduction rate of seroma in the drain group [24], but the previous systematic review and the meta-analysis includes studies conducted before 2004, and so its findings should be reassessed in light of 20 years of advancements in surgical techniques and technology. Similar results for the mean seroma volume were found by Soon et al.; they described a higher number of volumes in the no-drain group (856.7 mL) compared to the drain group (538.8 mL), but the differences were not checked to be statistically significant. In all cases, the reported volumes in patients without drainage refer to seromas that required drainage due to being symptomatic. However, there are few studies that compare the axillary volume in patients without drainage using ultrasound [26], showing a minimal volume after 15 days [27].

Despite these findings, some research suggests that the early removal of drains (within 24–48 h) does not significantly increase seroma rates and may facilitate faster recovery [18]. Furthermore, randomized trials have demonstrated that using additional techniques, such as axillary padding or flap fixation, can significantly reduce seroma formation, making drains unnecessary in select cases [17,28,29]. This padding or flap fixation consist of only three stitches between the skin subcutaneous flap and the pectoral and serratus fascia, with the objective of reducing the dead space.

### 4.3. Alternative Techniques to Reduce Seroma

Several authors studied the seroma after BC surgery. In 2009, Droeser and colleagues did not find a reduction in symptomatic seroma between the drain and no-drain groups [19]. They also included a group with sealant application and they observed a significant reduction in seroma formation associated with sealant use in patients who underwent mastectomy and ALND (*p* = 0.012). Other authors found similar results in seroma production using a new technique of suturing flaps without drains [30].

However, the previously quoted studies are outdated, and since then, the electrothermal bipolar vessel sealing system has emerged. These devices have demonstrated a significant reduction in seroma formation [31,32]. The surgical technique and technological advancements in surgical instruments also contribute to the progressive decrease in seroma formation, as they allow for better tissue control and preservation. Additionally, there has been an improvement in techniques with more conservative lymphadenectomies, now limited to Levels I and II, in contrast to the early 2000s, when Level III was also included.

To date, all these improvements contribute to a significant reduction in seroma formation compared to previous years. As a result, studies advocating no-drain approaches as a standard practice have started to emerge [26].

### 4.4. Recent Scientific Evidence

The Cochrane review, published in 2013, provides a comprehensive analysis of the use of drains in ALDN. However, it includes previously mentioned studies, with the most recent being Soon et al., published in 2005. The conclusion of this review is that the use of a drain does not eliminate the risk of developing seroma or the need for subsequent aspiration. While drains were associated with a lower incidence of seroma formation (OR 0.46, 95% CI 0.23–0.91, *p* = 0.03), they did not significantly reduce the total volume of seroma aspirated. Additionally, the presence of a drain did not impact infection rates (*p* = 0.14), but it was associated with a longer hospital stay (mean increase of 1.47 days, *p* = 0.0003). These findings suggest that while drains can help manage seroma formation, they do not fully prevent it, and alternative strategies should be considered in modern surgical practice [33].

Two recent studies have been found that include patients without an axillary drain: the GALA trial [26] and the REDHEMOPACH trial [34]. We will have to wait for the final results of GALA study to obtain more up-to-date conclusions.

The use of no drains impacts the short-term postoperative morbidity, such as seroma formation or drainage duration, and it does not seem to be related to disease recurrence or survival, but longer studies are needed to confirm this information.

Despite the lack of new comparative studies regarding the non-use of drains after ALDN, their routine placement remains common. This is primarily due to the fear of symptomatic seroma, which may require aspiration, or simply because it has traditionally been the standard practice. After an ALDN, surgeons must balance the comfort of not using drains for the patient against the possibility of needing aspiration in cases of symptomatic seroma. This decision ultimately leads to variations in clinical practice depending on the center and the treating surgeon.

In cases where drains are not placed, proper postoperative monitoring is essential, and symptoms should be managed effectively with analgesics, if needed, aiming to reduce the symptomatic seroma caused by pain. If aspiration is required, it should always be performed under sterile conditions to prevent secondary infections.

## 5. Use of Tissue Sealants in ALDN

With the aim of reducing seroma after ALDN, some authors have advocated the use of different tissue sealants either to close the axillary space or to decrease seroma formation. Depending of the type of sealant, it can either seal lymphatic vessels or facilitate tissue adhesion. Most articles specify that the axillary sealant should be placed after performing the axillary lymphadenectomy, checking for hemostasis and prior closure of the subcutaneous tissue and skin.

We find in the literature studies assessing various sealants, being either used alone or in conjunction with drains. Some authors suggest that combining drains with sealants may not offer the same advantages as using sealants alone, as drains could hinder the stabilization of the sealant in the lymphadenectomy bed and disrupt tissue adherence.

The main disadvantage of using sealants is the cost, but it should be evaluated whether their use reduces other types of costs, such hospitalization time, additional medical consulting, or avoiding the use of drains.

Several randomized controlled trials and observational studies have evaluated both the clinical and economic impacts of sealants in axillary surgery but the results of the studies are mixed and a comprehensive economic analysis was not carried out in the majority of them. Some authors have even reported that the use of certain sealants is not cost-effective [35]; therefore, this should be thoroughly analyzed in proper cost analyses.

The use of sealant influences short-term outcomes such as the seroma volume; therefore, no long-term results for recurrence or survival are demonstrated or analyzed.

### 5.1. Types of Sealants

#### 5.1.1. Fibrin-Based Sealants

Fibrin glue is one of the most commonly studied biological sealants and is composed of fibrinogen and thrombin, previously of bovine origin but now of human origin. Fibrin sealants function as biological hemostatic agents, interacting with tissues during surgery to support hemostasis, stimulate fibroblast growth and proliferation, and facilitate the closure of lymphatic vessels. These products are easy to apply, with rapid polymerization (60 s) [29]. Some examples of this fibrin sealants are Tisseel^®^, Evicel^®^, Artiss^®^, Tissucol, and others. This type of sealant is used in many surgical scenarios [36].

There are different randomized clinical studies that assess the outcomes after the application of fibrin sealants in patients undergoing either mastectomy or breast-conserving surgery. These studies conclude that the total volume of seroma production is lower with a fibrin sealant [37,38,39]. However, there are studies that do not show these differences in volumes but can be criticized for the small number of patients (30–40 patients) [39,40,41] or for their methodology [2].

Conversano et al. associate fibrin sealants with the closure process “padding22” described previously, and a reduction in the hospital stay duration was observed (*p* < 0.001), achieving a mean cost reduction per patient of EUR 2619 [29]. This study did not show differences in seroma formation or the number of punctures required. However, it has been previously described in those studies, including more recent ones, which do find volume differences in favor of the sealant group [42]. Nevertheless, this latter study evaluates the placement of a drain plus sealant, unlike the previous one where no drains were used. Therefore, drawing clear conclusions regarding the benefit of the sealant in patients with or without drainage may present a significant methodological bias.

In the most recent meta-analysis published by Chang et al. on the use of these sealants, some studies included patients who underwent mastectomy, which is already known to typically present with larger seroma volumes. In most of them, patients in both groups had drains, and it was described that in the fibrin sealant group, there was a lower seroma volume and shorter drainage duration, while the results regarding the incidence of symptomatic seroma, length of hospital stay, or infection remained the same [43].

#### 5.1.2. Polyethylene Glycol (PEG)-Based Sealants

Polyethylene glycol (PEG)-coated patches, such as Hemopatch™, have been introduced as an alternative to drains. Some studies have reported increased seroma formation but a reduction in emergency visits and outpatient aspirations in the intervention group. This discrepancy may be attributed to complications associated with drains rather than the effect of Hemopatch itself, as described in their findings [34].

#### 5.1.3. Cyanoacrylate-Based Sealants

Cyanoacrylate adhesives, such as Glubran 2^®^, are synthetic adhesives with rapid polymerization properties, creating a strong bond over tissue.

For this reason, cyanoacrylate has been used in various surgical procedures as an adhesive and for sealing structures. Among its approved uses is its application after inguinal or ALDN to reduce seroma formation.

There are studies suggesting that the use of cyanoacrylate sealants reduces seroma formation. However, most of these studies include patients undergoing mastectomy, which involves a larger dissection surface, and report decreases in the seroma volume and drainage duration [44]. Conversely, other studies have not found significant differences in seroma volumes in patients with both drainage and sealant application [41]. However, these studies often have a small sample size and methodological limitations that may influence their outcomes [35,45]. A recent study with a more robust methodological design also failed to demonstrate differences in seroma volumes following sealant application [45].

There are also studies that have found a higher seroma volume after the application of Glubran 2 [46].

In general, these studies include both patients undergoing breast-conserving surgery and mastectomy, all of whom receive drainage. To directly assess whether the sealant is associated with a reduction in seroma volume, studies with stricter selection criteria will be necessary to evaluate its effectiveness independently in mastectomy or lymphadenectomy. Additionally, it will be essential to analyze studies without drainage to determine whether its presence influences the sealant’s effectiveness. For this, we must wait for the results of the GALA study [26].

#### 5.1.4. Thrombin and Fibrinogen Combipatches

TachoSil^®^, a combination patch containing thrombin and fibrinogen, has been evaluated for its lymphostatic properties.

Studies comparing TachoSil with standard drainage techniques have yielded mixed results. While some research suggests that TachoSil may help reduce the hospital stay duration, it does not appear to significantly prevent seroma formation or decrease the volume of drained lymph fluid [47,48].

In Table 1 shows the different outcomes according to the placement of drainage or sealant.

## 6. Minimally Invasive and Endoscopic Surgical Approaches

Throughout the years, ALDN has been a key component in the staging of BC, with the majority of cases resulting in negative lymphadenectomies. After 2010, ALDN shifted from a staging procedure to a curative treatment, and is no longer performed in patients with luminal tumors and T1 or T2 stage tumors. Today, axillary lymphadenectomy is only indicated in cases of a high axillary burden, with more than three positive axillary lymph nodes, or in the context of persistent node positivity after neoadjuvant therapy. However, this indication remains a topic of ongoing debate in the current clinical landscape.

The conventional and classic technique for performing an ALDN is open surgery. However, minimally invasive or endoscopic surgery for performing endoscopic lymphadenectomies dates back to the 1990s [49]. These techniques emerged as alternatives to open surgery with the goal of reducing morbidity without compromising oncological efficacy. Most of the published studies on this topic date back to before 2010 and include a small number of cases.

There are more recent meta-analyses, published in 2020, that demonstrate no significant differences between the endoscopic or MALND (Mastoscopic Axillary Lymph Node Dissection) and conventional or CALND (Conventional Axillary Lymph Node Dissection) groups in terms of the number of lymph nodes removed, tumor recurrence rate, axillary drainage, postoperative hospitalization time, and tumor size [50], but most of the studies included were performed before 2017; therefore, the evidence of axillary endoscopic lymphadenectomy for BC should be re-evaluated in light of the new current surgical indications and technology.

### 6.1. Endoscopic Axillary Lymphadenectomy

In the early stages of endoscopic lymphadenectomy, some authors described the technique by combining lymphadenectomy with liposuction to facilitate axillary dissection [51,52]. However, liposuction presented several drawbacks, such as tissue fragmentation, poor preservation of lymph nodes, and can increase vascular trauma. For this reason, authors like Kamprath et al. [53] and Tagaya et al. [49] later described the technique without added liposuction, correctly identifying the vascular and nervous bundles. These latter authors (2002) achieved an adequate number of extracted lymph nodes and a lower incidence of neuromuscular complications.

Recently, in 2023, studies have emerged regarding the endoscopic sentinel lymph node detection, highlighting some benefits such as a lower seroma rate and improved aesthetic outcomes, although at the expense of a longer surgical time [54,55].

Several comparative studies have analyzed the benefits of endoscopic axillary lymphadenectomy in relation to the conventional technique. The systematic review conducted by Aponte-Rueda et al., which examines 12 studies with case series ranging from 1997 to 2007, describes a longer operative time in endoscopic surgery, with no differences in the quality of lymph node extraction (>10 nodes) and a low complication rate [56].

As we previously mentioned, the most recent meta-analysis published by Xiong et al. found that endoscopic surgery is associated with less intraoperative bleeding, a shorter drainage duration, and a lower incidence of postoperative complications, although with a longer surgical time [50]. 

De Wilde et al. (2003) and Chengyu Luo confirmed that endoscopic lymphadenectomy does not compromise the quality of axillary staging and reduces the incidence of lymphedema and sensory alterations in the arm [57,58], resulting in a better aesthetic appearance [58].

Seroma has also been found to be different between the two types of approaches, being lower in endoscopic surgery compared to conventional surgery (10 vs. 45%) [53] and in endoscopic BSGC [55].

The surgical time, which is described as longer than classical surgery, can be reduced after experience. Kamprath et al. reported that the first 16 patients had significantly longer operative times, but after 17 cases, the surgical time was reduced, suggesting that experience and training play a crucial role [53]. It is suggested that with adequate training, a surgeon may become proficient in this technique after performing a limited number of cases. Now, the endoscopic technique has been performed and modified by new technological advancements (Figure 1). The injection of the colorant indocyanine green in the areola helps to identify all the lymph nodes from the territory of the mammary gland. This method has been used for lymph node detection [59] and also in endoscopic ALDN to facilitate the identification of all the axillary lymph nodes easily seen in green (Figure 1).

In terms of oncological outcomes, the previously described studies reported an axillary recurrence rate of 0.5%, with no significant differences between groups regarding local recurrence, either in the axilla or the breast. Regarding the number of lymph nodes, all studies demonstrated the removal of more than 10 lymph nodes [56,60].

### 6.2. Robotics and the New Frontier of Axillary Surgery

To date, robot-assisted ALDN for the treatment of BC is not widely approved and remains limited to research scenarios. In regions such as Europe and Asia, studies suggest that robotic surgery for mastectomy may offer advantages, including smaller incisions and improved aesthetic outcomes. However, it , is still being evaluated in clinical trials.

More recently, robotics has revolutionized minimally invasive surgery, offering additional benefits in precision and ergonomics. Studies on the da Vinci system have shown that robot-assisted ALDN can be a safe and effective procedure, reducing surgical trauma while enhancing cosmetic results. Despite its potential, further clinical validation is required before robotic axillary lymphadenectomy becomes a standard practice in BC treatment.

Only a few studies have been found on robot-assisted ALDN, as it must be performed under clinical trials and in centers with prior experience in robotic breast surgery, as previously mentioned [61,62]. These studies have been conducted mainly in Korea and China, where the experience with robotic surgery is higher.

Both studies were associated with mastectomy, but Chen et al. also incorporated axillary lipolysis during the same surgical procedure.

The rate of infection, fat necrosis, and lymphedema (6.67% vs. 26.67%) was lower with the robotic technique according to Chen et al., with no significant differences in other parameters such as bleeding, surgical time, or recurrence. A later study performed in Korea [61] also found no differences between the two techniques in terms of perioperative outcomes, except for smaller incision sizes in the robotic approach. The study further highlights that since robotic surgery may require a longer operative time, technical training is essential to develop adequate surgical skills.

Regarding robot-assisted endoscopic lymphadenectomy, we are still in the early stages of the technique. In fact, the introduction of the da Vinci Single Port system may add an advanced technology with respect to the classical one in axillary dissection due to the limited space in the cavity. However, due to its high cost, this single port robot is currently available in a limited number of centers.

In conclusion, robot-assisted ALDN is in the research and development phase and has not yet been approved as a routine technique for clinical practice in the surgical treatment of breast cancer. Its application is limited mainly to centers with extensive experience in robotic surgery and within the framework of research protocols or controlled clinical trials. Therefore, although preliminary studies suggest potential benefits in terms of reduced morbidity and improved cosmetic outcomes, robust evidence is still required through larger clinical trials with longer follow-up to support its oncological safety, efficacy, and cost-effectiveness before its widespread adoption.

New technologies also bring potential negative outcomes, such as longer surgical times and the need for more resources and specialized training that cannot be provided in all centers, except in the reference centers. This fact can hinder the expansion of these practices to all centers equally.

## 7. Breast Cancer Axillary Lymphadenectomy and Quality of Life

### 7.1. Introduction

In developed countries, most breast cancer patients survive for long periods of time, with an overall survival at five years of 91.7% (SEER 21). Surgery remains the only curative treatment for breast cancer. Nevertheless, the improvement in survival is mainly due to the development of systemic treatments and the early diagnosis of cancer by screening programs. In this context, surgeons have adjusted the aggressiveness of surgery to reduce the post-surgical sequelae. This means that new objectives are becoming increasingly important, such as the assessment of quality of life, health status, and cosmetic impact of long-term survivors. Psychological resilience and social support significantly influence recovery and QoL, potentially moderating the perceived impact of surgical morbidity.

Patient-reported outcomes (PROs) help to reduce the gap between the information provided by clinicians and the patient’s own experience, leading to better communication and improved information on the care provided to patients. In addition, they allow the selection of treatments that provide the most benefit to patients, not only in terms of survival or a reduction in recurrence but also by including the valuable patient choice, which clinicians may not fully perceive. PROs are defined as any information about the state of the patient’s health condition that comes directly from the patient (or in some cases from the patient’s caregiver), without an interpretation of the response by healthcare personnel or any other person [63].

### 7.2. Conservative Surgery of the Axilla

The ACOSOG Z0011 [64,65] and AMAROS [66] studies have produced a paradigm shift in axillary surgery in breast cancer patients: the possibility of avoiding axillary lymph nodes dissection/lymphadenectomy (ALND) in a specific group of patients with initial axillary node involvement. Omitting lymphadenectomy brings benefits to patients, especially by reducing the morbidity associated with the surgical technique, mainly lymphedema.

In recent years, the ACOSOG Z1071 [67], SENTINA [68], and TSN FNAC [69] trials have observed the possibility of avoiding lymphadenectomy in cN1 patients who were treated with primary systemic treatment (PST) and presented negativization of the initial axillary involvement. In order to assess the effect of PTS on axillary disease, the MD Anderson group [70] designed the targeted axillary dissection (TAD) that provides the possibility of evaluating the pathological response of axillary disease after PST, with an acceptable false negative rate (FNR) (<10%). TAD includes the resection of the affected node and the resection of three or more sentinel nodes.

We suppose that TAD and sentinel node biopsy (SNB) reduces the impact of surgery on patients with breast cancer, but what the real impact does this de-escalation have on patients’ quality of life?

### 7.3. Quality of Life and Breast Cancer Surgery

When we analyze the quality of life (QoL) of breast cancer patients, we must consider that surgical treatment usually includes two associated aims, local control of the cancer and axillary staging. The QoL of surgical breast cancer patients is influenced by the surgery performed on the breast and on the axilla. In addition, QoL is also influenced by radiotherapy and associated systemic treatment.

Conservative surgery seems to have a lower overall impact on patients’ QoL [71,72]. Recent studies show that reconstruction after mastectomy can offer a comparable benefit in some dimensions such as satisfaction with the breasts and physical well-being [73]. The type of surgery performed on the breast also has a negative effect on the ipsilateral arm symptoms. Boechmer observed five times more arm symptomatology in patients with mastectomy, with or without reconstruction, in the regression model with respect to conservative surgery [74]; Laws observed greater arm symptomatology in patients with mastectomy + reconstruction, especially in autologous reconstruction [75]; and Enien notes greater arm symptomatology in patients with modified radical mastectomy than in patients with conservative surgery [76]. The IDEAL study also observed greater arm symptomatology in patients with mastectomy, with an independent effect of surgery adjusted for radiotherapy [77]. However, in the UK START trial on the effect of radiotherapy on the quality of life of surgical patients, patients with conservative surgery had a worse quality of life than patients with mastectomy [78]. It concluded that the most influential factors in these patients were chemotherapy and the age of the patients.

Regarding axillary surgery, the situation is complicated, since in pN1 patients in whom lymphadenectomy is obviated, axillary radiotherapy (RT) is administered with a therapeutic intent. We know that RT itself has an impact on patients’ quality of life and on arm-related symptoms. The main studies assessing the effect of ALND on breast cancer patients focus on the effect of this technique vs. SNB (Table 2). Most of them did not use specific tools/questionnaires to assess the morbidity of the affected arm and used subscales of general questionnaires such as the QLQ-BR23 that include 2–3 specific questions about arm involvement.

Barranger observed greater symptomatology in the ipsilateral arm of patients with ALND but found no significant changes in the quality of life of patients (measured by QLQ-c30 and QLQ-BR23) and no significant differences in difficulties at work and/or with daily activities [83]. The 10-year review of the study AMAROS observed that 44.2% of the patients reported lymphedema at any time point after ALND plus RT compared with 28.6% of the patients after SNB plus RT [84]. For two items of the arm symptoms scale, a statistically significant difference was observed. For QoL, no statistical differences were observed between any of the selected scales (arm symptoms, pain, or body image) at 1, 3, and 5 years after treatment. Belmont has shown that the relevant differences persisted between SNB and ALND groups for arm morbidity and its impact on functioning and QOL throughout the 1-year period after treatment [80]. An arm volume change corrected for contralateral change was more frequent in patients with ALND (51.1% vs. 14.5% *p* = 0.062). However, the difference in QoL between groups, measured by generic QoL questionnaires (SF36 and FACT-B+4), was negligible. Dabakuyo, in a multicenter cohort study, analyzed the QoL of patients undergoing ALND vs. SNB according to surgeon preference. QoL was evaluated using the EORTC QLQ-C30 and the EORTC QLQ-BR-23 to assess the global health status (GHS), and the arm (BRAS) and breast (BRBS) symptom scales. They observed that the lowest BRAS QoL scores were recorded in the SNB group, with no changes in the GHS and breast (BRBS) symptom scale scores [81]. The randomized trial by Del Bianco et al. observed that the SNB group had significantly less lymphedema, movement restrictions, pain and numbness with respect to the ALND group [79]. The mean scores of the Psychological General Well-Being Index (PGWB) were significantly better in the SNB group than in the ALND group, but this difference was no longer significant at 24 months. Kootstra studied the morbidity related to axillary surgery [85]. Significant time effects were found on the difference scores for the four functional shoulder measurements. They observed that limitations in shoulder function and arm strength were most severe immediately after surgery for breast cancer, and by the 2-year follow-up, all women had regained their preoperative level of arm muscle strength. Women in the ALND group showed a gradual progression of lymphedema during the 2 years of follow-up, whereas the women in the SNB group did not develop lymphedema. Al Nakib observed greater arm symptomatology in patients with ALND. He did not observe significant differences in overall QoL (QLQ-BR23) but the global quality of life score was significantly correlated with the pain level and the arm or breast scores [82]. Rietman observed that treatment-related upper limb morbidity, perceived disabilities in ADL, and worsening of QoL at 2 years after surgery were significantly less after SNB compared to ALND [86].

In summary, most of the studies that investigated the impact of lymphadenectomy on quality of life use tools to assess the general quality of life of breast cancer surgical patients. The use of specific probes or questionnaires focused on the morbidity of surgery in the ipsilateral arm is less frequent. It seems clear that axillary lymphadenectomy increases the frequency of arm symptoms and sequelae such as pain or lymphedema when compared to SNB, especially when associated with RT. These differences appear to be reduced over time. Despite the sequelae, in general quality of life questionnaires, such as the QLQ-30 or QLQ-BR23, most studies do not observe a significant impact of the morbidity of axillary lymphadenectomy on the overall quality of life of patients. Some authors interpret this result as being due to the effect of systemic treatments, especially chemotherapy and hormone therapy, on patients’ quality of life, which probably dampens the effect of surgery on overall quality of life. In addition, as we have seen above, the breast surgery itself is also a factor that affects quality of life and overlaps with the effect of axillary surgery. On the other hand, it is common when studying the quality of life of cancer patients to observe these contradictory phenomena. Physicians measure the effects of treatments that they believe affect patients. Quality of life studies inform us about the real impact that these treatments have on patients.

## 8. Discussion

The evolution of ALDN in BC surgery has been marked by significant advancements in surgical techniques, perioperative care, and the optimization of patient outcomes. Our analysis highlights key aspects that influence the practice of ALDN, including ambulatory surgery feasibility, the role of surgical drains, the effectiveness of sealants, the development of minimally invasive approaches, and QoL.

### 8.1. Ambulatory ALDN

Ambulatory ALDN has emerged as a safe and cost-effective strategy that enhances patient autonomy, reduces hospital-related complications, and optimizes healthcare resources. ERAS protocols have played a pivotal role in facilitating early discharge, demonstrating that patients can achieve similar, if not superior, outcomes in the ambulatory setting compared to traditional inpatient care. However, challenges remain in ensuring that patient selection criteria are appropriately defined to maximize safety. Future research should focus on refining surgical techniques to further reduce seroma formation and ensure optimal recovery while maintaining the safety standards of outpatient surgery.

### 8.2. Use of Drains

The role of drains in ALDN continues to be debated, with studies demonstrating both benefits and drawbacks. While their primary function is to reduce seroma formation and the need for symptomatic aspirations, evidence suggests that their routine use may not be justified. The Cochrane review from 2013 concluded that although drains decrease the seroma incidence, they do not eliminate it entirely nor reduce the need for aspiration. Moreover, drains do not impact infection rates but do contribute to longer hospital stays and potential discomfort at home, impacting patients’ QoL. Older studies from the late 20th century supported the necessity of drains, yet more recent findings challenge this notion, particularly with the advent of advanced surgical tools such as the electrothermal bipolar vessel sealing system and less extensive lymphadenectomy techniques. Given these developments, modern research increasingly supports no-drain approaches, aligning with efforts to improve postoperative recovery and minimize complications.

### 8.3. Use of Sealants

Sealants represent another important factor in ALDN, with studies suggesting they may play a role in reducing seroma formation. However, inconsistencies in study designs—particularly the inclusion of diverse patient populations undergoing varying types of breast surgery—have led to conflicting conclusions. Patients undergoing mastectomy have larger dissection areas, which inherently influence seroma formation, and the presence of a drain may interfere with sealant effectiveness. Among the available options, fibrin-based sealants appear to have the strongest supporting evidence. Given the increasing movement toward omitting drains, future studies should specifically assess the effectiveness of different sealants in patients managed without drainage. The upcoming results from the GALA study may provide further insights into the independent role of some sealants in seroma prevention.

### 8.4. Minimally Invasive ALDN

Minimally invasive ALDN has demonstrated promising outcomes in reducing surgical complications while improving functional and aesthetic results. Endoscopic and robotic techniques have gained traction, with the evidence suggesting comparable oncological safety and complication rates to conventional surgery. Additionally, these minimally invasive approaches have been associated with a lower incidence of sensory disturbances and seroma formation, likely due to reduced tissue trauma and better neurovascular preservation. However, challenges persist, including the need for specialized training and the learning curve associated with endoscopic approaches within the confined axillary space. While the relatively straightforward anatomy of the axilla may facilitate skill acquisition, widespread adoption will require high-quality multicenter studies to validate these findings. Furthermore, robotic-assisted lymphadenectomy, while promising, remains in the experimental phase and demands thorough cost analyses to determine its practicality for broader clinical application. Factors such as accessibility, cost-effectiveness, and technical expertise must be carefully considered before these approaches can be widely integrated into routine surgical practice.

### 8.5. Quality of Life and ALDN

Quality of life (QoL) considerations have become increasingly critical in the modern surgical management of breast cancer patients undergoing axillary lymphadenectomy. Although ALND is associated with higher rates of arm morbidity, including pain, reduced mobility, and lymphedema, compared to sentinel node biopsy, its overall impact on global QoL scores appears surprisingly limited. This paradox may be explained by the overlapping influences of systemic therapies and breast surgery on patients’ overall health perception. Nevertheless, minimizing surgical trauma through techniques such as minimally invasive lymphadenectomy, early drain removal, and outpatient surgery models has shown promise in preserving functional outcomes and improving postoperative recovery. As the field continues to evolve, integrating patient-reported outcome measures (PROMs) into routine clinical practice will be essential to better understand and address the real-world QoL impact of axillary surgery, ensuring that oncological safety is achieved without sacrificing the well-being of long-term breast cancer survivors.

### 8.6. Clinical Considerations

The current review does not aim to provide definitive clinical guidelines or recommendations; however, it does aim to show readers the different studies on the major topics of discussion surrounding ALDN in order to provide information and criticize the current management by surgeons. Due to the disparate results in some areas, such as those related to sealants, a critical opinion on the matter must continue to be maintained.

We can conclude that outpatient surgery has proven feasible and, therefore, thanks to appropriate protocols, patient education, and close follow-up systems, can be applied with clear benefits.

Regarding the omission of drains, it has been shown that their placement by the surgeon is still a routine common practice. There are multiple disparate articles on the subject with heterogeneous data collected. The technological advances in sealing achieve acceptable rates of symptomatic seroma without drainage. The no-drain replacement could be recommended, as it improves autonomy for the patient and reduces the rate of emergency department return visits due to drain malfunction or infection.

The use of sealants remains controversial in a clinical and relevant way. Fibrin sealants appear to achieve a greater volume reduction; however, the clinical importance is the percentage of seroma that is clinically relevant and is this the one that should be taken into greater consideration (symptomatic seroma). Due to the variability and heterogeneity of studies (with or without drainage, without sealant, mastectomy, etc.), it is difficult to reach conclusions regarding their use, specifically which type of sealant. If we have demonstrated that drainage provides no benefits, studies of sealants in patients without drainage are probably necessary, as is not frequent in the literature.

Minimally invasive surgery has demonstrated benefits in clinical practice, and ALDN is no different. We recommend that surgeons learn and train in minimally invasive surgery. Furthermore, ongoing prospective studies are necessary to establish stronger recommendations.

## 9. Limitations

The studies reviewed are heterogeneous, with different series that, in some cases, include conservative surgeries with lymphadenectomy or patients undergoing mastectomy. This variability can lead to differing conclusions, as mastectomy generally results in higher seroma production and almost always requires drainage placement. To avoid the effect of mastectomy on the interpretation of the findings, priority was given to studies focusing on conservative surgery with lymphadenectomy, excluding findings from mastectomy patients. In consequence, the conclusions of this review could not be directly applied to patients subjected to mastectomy.

A limitation of the present review is the inclusion of studies with small, heterogeneous sample sizes and lack of long-term outcome data in some cases.

## 10. Future Directions

As the surgical management of axillary disease in breast cancer continues to evolve, several key areas warrant further investigation. Future studies should focus on optimizing patient selection criteria for ambulatory ALND, including standardized protocols to ensure safe same-day discharge. High-quality randomized trials are needed to better define the role of drains, particularly in light of new surgical technologies and minimally invasive approaches that may reduce seroma formation independently. The impact of tissue sealants on seroma prevention also requires clarification through well-designed studies that specifically separate outcomes by surgical technique and the presence or absence of drainage. Furthermore, while early data on minimally invasive and robotic ALND are promising, larger prospective trials are necessary to confirm their oncological safety, cost-effectiveness, and long-term benefits in terms of functional recovery and QoL. The implementation of these research strategies may face challenges, including variability in healthcare infrastructure, differences in clinical training, and unequal access to minimally invasive technologies across regions

As patient-centered care becomes increasingly prioritized, future research must also incorporate patient-reported outcome measures (PROMs) to better capture the real-world impact of surgical innovations on quality of life. Finally, ongoing technological advances, including the refinement of robotic platforms and the development of enhanced sealing devices, offer exciting opportunities to further minimize morbidity and improve the outcomes and QoL of breast cancer patients requiring axillary surgery.

## 11. Conclusions

Axillary lymphadenectomy in breast cancer surgery continues to advance, with ongoing efforts aimed at improving patient outcomes through innovation in surgical techniques and perioperative management. The transition toward ambulatory surgery, the reconsideration of drainage necessity, the exploration of sealants, and the development of minimally invasive techniques all represent key advancements in the field. Although axillary lymphadenectomy increases arm morbidity compared to sentinel node biopsy, its overall impact on quality of life appears limited, likely due to the overlapping effects of systemic therapies and breast surgery.

Future research should focus on refining these strategies to further enhance surgical safety, reduce complications, and improve the overall QoL for patients undergoing axillary lymphadenectomy.

## Figures and Tables

**Figure 1 cancers-17-02201-f001:**
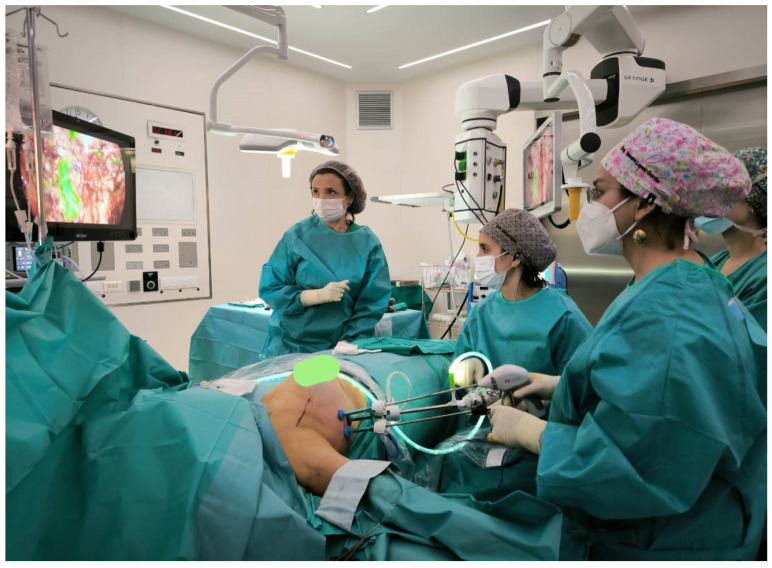
Endoscopic axillary lymphadenectomy.

**Table 1 cancers-17-02201-t001:** Comparative outcomes: sealants vs. drainage.

Outcome	Drainage	Sealants
Seroma formation	Common	Variable (higher in some studies, lower in others)
Hospital stay	Prolonged	Reduced in some sealant groups
Need for postoperative aspirations	Frequent	Reduced in some sealant groups
Pain and discomfort	Higher due to drains	Lower in most sealant groups
Cost-effectiveness	Increased outpatient visits	Potential cost savings

**Table 2 cancers-17-02201-t002:** Studies on QoL after axillary lymphadenectomy for breast cancer.

Study/Year	Patients, N	Pain	Arm Swelling	Dysesthesias	QoL
Barranger E et al., 2005 [79]	SNB, n = 54	21.2%	0%	5.7%	7.6
	ALND, n = 61	52.9%	21.6%	51%	7.6
	SNB + ALND, n = 10	60%	10%	50%	7.7
Belmonte et al., 2012 [80]	SNB, n = 64	ND	11.8%	9.8%	119.05
	ALND, n = 29		35.5%	69.2%	111.89
Dabakuyo et al., 2009 [81]	SNB, n = 222	ND	ND	ND	75.9
	ALND, n = 235				74.5
	SNB + ALND, n = 61				68.0
Al Nakib et al., 2010 [82]	SNB, n = 212	39.4%	9.9%	40.1%	73.6
	ALND, n = 131	67.7%	33.6%	83.1%	68.8

ND, no data; SNB, sentinel node biopsy; ALND, axillary lymph node dissection; n, number; QoL, quality of life.

## Data Availability

No new data were created or analyzed in this study. Data sharing is not applicable to this article.

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
