# Peer review of "Breaking Dogmas in Axillary Lymphadenectomy and Quality of Life"

_cancers, 2025, doi:10.3390/cancers17132201_

Round 1
Reviewer 1 Report
Comments and Suggestions for Authors
Associated with the surgical treatment of breast cancer, axillary lymph node chains are removed as a method of diagnosing cancer invasion. However, 70% of cases result in the development of seroma or lymphedema. Lymphedema is the accumulation of fluid in the dead space created by the removal of lymphoid tissue, which causes patient discomfort, an increased risk of infection, and delays adjuvant therapy. This review compares various modifications and innovations with the classic technique. It addresses outpatient surgery, the placement of sealants and drains, and minimally invasive surgery, along with a brief review of the quality of life of these patients. It addresses historical aspects, the statistics of classic techniques, and why they seem to resist modification over the years. This is a well-structured article that will allow the specialized sector to review the history and, above all, assess the quality of life of patients. It is an important contribution to considering the incorporation of novel and less invasive techniques.
Author Response
We appreciate the reviewer's assessment and hope that the information included in this review will help readers consider new surgical scenarios regarding axillary lymphadenectomy for breast cancer.Reviewer 2 Report
Comments and Suggestions for Authors This review has some scientific value. However, I have the following concerns: The authors mentioned that they conducted a systematic keyword-based literature search across Scopus, PubMed, Web of Science, and Google Scholar, but they did not specify the exact time range of the search. The article states that ALND is becoming less necessary in breast cancer surgery but is still required in certain cases, such as those with high axillary tumor burden. However, the authors did not elaborate on how to accurately assess the axillary tumor burden. For example, is it judged through imaging examinations (such as ultrasound, MRI, etc.) or in combination with clinical examination and other biomarkers? Different assessment methods may lead to differences in the judgment of axillary tumor burden, which in turn may affect the indication for ALND. When discussing ambulatory surgery and early discharge, the authors mentioned that patient beliefs are a major factor hindering early discharge. However, the article does not provide specific strategies for changing patients’ beliefs. The article discusses the controversy surrounding the use of drains. Although it mentions that omitting drains may increase seroma volume, it does not delve into whether this increase in seroma volume will have a significant impact on patients’ long-term prognosis. Moreover, for patients with drains, the article does not provide detailed information on drain care measures, such as methods for fixing the drain, observation and recording of drainage fluid, etc., which can also affect patients’ recovery. In the section on tissue sealants, the authors mentioned several types of sealants, including fibrin-based sealants, PEG-based sealants, cyanoacrylate-based sealants, and thrombin and fibrinogen combipatches. However, the article does not specify the optimal use of these sealants in different surgical scenarios. For example, should fibrin-based sealants be applied immediately after a specific surgical step or before closing the incision? In addition, the cost-effectiveness analysis of sealants is only briefly mentioned, without a detailed calculation considering the overall cost savings from reduced hospital stay and avoidance of drains. The article does not consider the impact of patients’ psychological factors and social support systems on quality of life, which may to some extent mask the effect of ALND on quality of life. The article proposes future research directions, including optimizing patient selection criteria for ambulatory surgery and clarifying the role of drains. However, the authors do not mention the potential research barriers. For example, in optimizing patient selection criteria for ambulatory surgery, there may be challenges such as different healthcare systems and uneven distribution of medical resources in different regions. Will these factors affect the implementation of the research? The authors are advised to address the above issues to improve the quality of the paper.Author Response
Comment 1: The authors mentioned that they conducted a systematic keyword-based literature search across Scopus, PubMed, Web of Science, and Google Scholar, but they did not specify the exact time range of the search.
Response 1: We have added the following sentence in the method section: “covering publications from January 1980 to December 2024”
Comment 2: However, the authors did not elaborate on how to accurately assess the axillary tumor burden. For example, is it judged through imaging examinations (such as ultrasound, MRI, etc.) or in combination with clinical examination and other biomarkers? Different assessment methods may lead to differences in the judgment of axillary tumor burden, which in turn may affect the indication for ALND.
Response 2: We totally agree with the reviewer and we add a sentence to explain tumor burden in the introduction section as follows: Axillary tumor burden is generally assessed through a combination of clinical examination, imaging modalities such as ultrasound and MRI, and a histopathological confirmation is always required before the indication of performing an axillary lymphadenectomy.
Comment 3: When discussing ambulatory surgery and early discharge, the authors mentioned that patient beliefs are a major factor hindering early discharge. However, the article does not provide specific strategies for changing patients’ beliefs.
Response 3: Some surgical procedures that traditional required hospitalization has shifted to early discharge regimes. In most cases, this has been achieved thanks to improve patient’s information about early discharge, and it has required time for patient acceptance and improve workflow management. The information patients must receive is basic and essential. Therefore, thanks to the reviewer’s recommendation, we have added a sentence explaining strategies to improve patient acceptance for early discharges in the Ambulatory Surgery and Early Discharge Post ALND section.
“The role of case management nurses and the surgeons' dedication to explaining outpatient discharge pathways, preoperative education and shared decision-making approaches can improve patient confidence and thus facilitate the acceptance of early discharge”.
Comment 4: The article discusses the controversy surrounding the use of drains. Although it mentions that omitting drains may increase seroma volume, it does not delve into whether this increase in seroma volume will have a significant impact on patients’ long-term prognosis.
Response 4: We appreciate this observation. We have now added a discussion noting that while increased seroma volume can impact short-term recovery and comfort, there is limited evidence suggesting it significantly affects long-term prognosis. We add in the section 4.2 Use of Drains the following sentence: “While larger seroma volumes may prolong recovery and cause discomfort, current literature does not strongly support a link between seroma volume and long-term oncological outcomes”
Comment 5: Moreover, for patients with drains, the article does not provide detailed information on drain care measures, such as methods for fixing the drain, observation and recording of drainage fluid, etc., which can also affect patients’ recovery.
Response 5: We agree with the reviewer and we have included a brief description of standard drain care measures such as fixation, output monitoring, and signs of complications that could explain better the patient’s recovery. We add the sentence in 4.1 types of Drainage Protocols section: “Standard drain care includes securing the drain to prevent dislodgement, daily monitoring of fluid output, and vigilance for signs of infection or blockage”
Comment 6: However, the article does not specify the optimal use of these sealants in different surgical scenarios. For example, should fibrin-based sealants be applied immediately after a specific surgical step or before closing the incision?
Response 6: We consider the information requested by the reviewer to be relevant, and therefore have added a sentence in 5 Use of Tissue Sealants in ALDN section: “ Most articles specify that the axillary sealant should be placed after performing the axillary lymphadenectomy, checking for hemostasis and prior closure of the subcutaneous tissue and skin”.
Comment 7: In addition, the cost-effectiveness analysis of sealants is only briefly mentioned, without a detailed calculation considering the overall cost savings from reduced hospital stay and avoidance of drains.
Response 7: Cost effectiveness analysis of sealant has not been carried out or is not deeply analyzed in the majority of the studies and it differs depending on the sealant used. According to the reviewer recommendation, we add a paragraph explaining the cost analysis studies information about sealants in 5 Use of Tissue Sealants in ALD: Several randomized controlled trials and observational studies have evaluated both the clinical and economic impact of sealants in axillary surgery but the results of the studies are mixed and a comprehensive economic analysis as not carried out in the majority of them. Some authors have even reported that the use of certain sealants is not cost effective [47] , therefore this should be thoroughly analyzed in properly cost analyses studies.
Comment 8: The article does not consider the impact of patients’ psychological factors and social support systems on quality of life, which may to some extent mask the effect of ALND on quality of life.
Response 8: Thank you for the important consideration. We have added content recognizing the role of psychosocial factors and support systems in influencing patient-reported quality of life. The current review aims to provide a general overview of quality of life and not go into specific details of all the variables included in quality of life related to LA. The psychological aspect is included in the questionnaires analyzed in this review (EORTC QLQ-C30, QLQ-BR23, FACT-B and SF-36). We add an extra sentence in 7.1 Introduction section: Psychological resilience and social support significantly influence recovery and QoL, potentially moderating the perceived impact of surgical morbidity”.
Comment 9: The article proposes future research directions, including optimizing patient selection criteria for ambulatory surgery and clarifying the role of drains. However, the authors do not mention the potential research barriers. For example, in optimizing patient selection criteria for ambulatory surgery, there may be challenges such as different healthcare systems and uneven distribution of medical resources in different regions. Will these factors affect the implementation of the research?
Response 9: This is a valuable point. We have now discussed potential implementation barriers such as healthcare disparities, resource availability, and institutional variation in practice. We add this information in 11 Future Directions section: “Implementation of these research strategies may face challenges, including variability in healthcare infrastructure, differences in clinical training, and unequal access to minimally invasive technologies across regions”.
We sincerely thank the reviewer for the comprehensive feedback, which has significantly improved the depth and clarity of our manuscript. We hope that the revised version addresses all concerns adequately.

Reviewer 3 Report
Comments and Suggestions for Authors
Authors should be congratulated for their effort to study this intriguing subject, which is quite difficult to approach. Axillary lymphadenectomy (ALND) is considered as a gold standard adjunct surgical procedure in some patients affected by breast cancer. Seroma formation is an expected postoperative complication causing an increased risk of infection, along with delays in adjuvant therapy compromising patients’ well-being. Surgical draining is the classic approach to reduce seroma volume promoting at the same time tissue adherence. Recently, the advent of new approaches (e.g. early drain removal, no-drain techniques, fibrin seals application, axillary padding, minimally invasive surgeries) seems to challenge this well-established practice. Postoperative arm dysfunction, pain and lymphedema are usual complaints secondary to ALND affecting quality of life (QoL). Subsequently, it is important to overview the impact of ALND, sealants, drains application, minimal invasive surgeries in QoL.
Authors in their material and methods section thoroughly presented every step of their searching methodology, followed by the new “trend” of ambulatory surgery and early post ALND discharge associated with a detailed analysis of their benefits and their economic parameters.
In the ensuing sections all hot topics (draining use, tissue sealants application with their various types, endoscopic and minimally invasive techniques) are thoroughly considered ending with a “taste” of the future with robotics whose utilization in this type of surgery is highly promising although demanding more studies. QoL is contemplated last based on the analysis of various studies (Barranger et al, Belmote et al, Debakuyo et al, and Al Nakib et al). It was extracted that most of them did not focus on the specific morbidity caused by ALND in the arm. They rather assessed general QoL. Subsequently, ALND did not seem to have a negative impact on the overall patients’ quality of life. Perhaps adjuvant therapies could contribute to it, overlapping the real effect of axillary surgery. Finally, in the discussion section all issues that have already been developed are highlighted.
Generally, it is a well-written manuscript that deserves to be published.
Author Response
We appreciate and thank the reviewer for the time he took to critically read this review. We can see that the topics covered are considered of interest to the scientific community to which they are addressed.Reviewer 4 Report
Comments and Suggestions for Authors
This is a nice review of a topic of interest to breast cancer surgery clinicians and survivors. The focus on how different lymphadenectomy techniques impact patient quality of life is innovative and important. It is also very important from a health policy and hospital operations perspective that the authors have investigated the potential cost savings of less invasive techniques that lead to fewer (or zero) inpatient days. The article is well written, and should provide pause for surgeons and other clinicians regarding their practices. I enjoyed reading the article and appreciate the authors' thoroughness.
The graphic is unnecessary and doesn't really make any sense. This can be eliminated.
Author Response
Many thanks to the reviewer for their constructive criticism of this manuscript. Minimally invasive surgery is undoubtedly present today and may increase in the near future. This fact, as the reviewer points out, will facilitate early discharge and better resource management. We are pleased to read that the reviewer enjoyed reading the article. Regarding the graphical abstract, it is up to the journal editors, as it was requested in the application. If the editors deem it appropriate, it may be removed.Round 2
Reviewer 2 Report
Comments and Suggestions for Authors
Although the paper mentions a variety of techniques and methods related to axillary lymph node dissection (ALND), such as minimally invasive surgery and the use of tissue sealants, the discussion of some emerging technologies, like certain novel robot-assisted surgical techniques, is not in-depth enough. The paper covers multiple topics, including day surgery, the use of drains, and the effectiveness of tissue sealants, but falls short in integrating these data for a comprehensive analysis. When discussing the advantages of some new technologies, the paper may overemphasize the positive aspects while neglecting potential negative outcomes. The paper discusses various innovative methods for ALND but fails to provide clear, evidence-based clinical recommendations. When discussing different surgical methods and postoperative care strategies, the paper does not fully consider their impact on patients' long-term prognosis. The paper's chapter divisions may not be clear enough, with some content overlap between sections. There may be issues with logical coherence when presenting viewpoints and analyzing data.
Author Response
Comment 1: Although the paper mentions a variety of techniques and methods related to axillary lymph node dissection (ALND), such as minimally invasive surgery and the use of tissue sealants, the discussion of some emerging technologies, like certain novel robot-assisted surgical techniques, is not in-depth enough.
Response 1:
We fully agree that robot-assisted axillary lymph node dissection represents a promising but still emerging area but this surgical indication still not included in the official manuals of robot-assisted, not even in da Vinci. All the studies related to robotic ALDN have to be under clinical trials approved by ethics committees. For this reason, until now, there has not been a further expansion of this surgical technique in which we can go deeper. In response, we have expanded the discussion to better address the current evidence on robotic approaches.
We add the following information in 6.2 section:
In conclusion, robot-assisted ALDN is in the research and development phase and has not yet been approved as a routine technique for clinical practice in the surgical treatment of breast cancer. Its application is limited mainly to centers with extensive experience in robotic surgery and within the framework of research protocols or controlled clinical trials. Therefore, although preliminary studies suggest potential benefits in terms of reduced morbidity and improved cosmetic outcomes, robust evidence is still required, through larger clinical trials with longer follow-up, to support its oncological safety, efficacy, and cost-effectiveness before its widespread adoption.
Comment 2: The paper covers multiple topics, including day surgery, the use of drains, and the effectiveness of tissue sealants, but falls short in integrating these data for a comprehensive analysis.
Response 2:
For greater clarification and to facilitate a structure that allows for easier reading of the manuscript, we have added the corresponding sections in the discussion.
Comment 3: When discussing the advantages of some new technologies, the paper may overemphasize the positive aspects while neglecting potential negative outcomes.
Response 3: Thanks to the reviewer for this observation, we add the potentials negative outcomes of the new technologies in 6.2 section and also some extra information in section 9 (limitations) as follows:
Section 6.2:New technologies also bring potential negative outcomes, such as longer surgical times, the need for more resources and specialized training that cannot be provided in all centers, except in the reference centers. This fact can hinder the expansion of these practices to all centers equally.
Section 9:
A limitation of the present review is the inclusion of studies with small, heterogeneous sample sizes and lack of long-term outcomes data in some cases.
Comment 4: The paper discusses various innovative methods for ALND but fails to provide clear, evidence-based clinical recommendations.
Response 4: We recognize the importance of offering practical guidance for clinicians. As such, we have added a new subsection in the Discussion titled "Clinical Considerations" in discussion section num 8, that summarizes potential recommendations based on the current evidence while stressing that these should be interpreted with caution due to the heterogeneity and limitations of existing studies.
Paragraph added:
Clinical considerations
The current review does not aim to provide definitive clinical guidelines or recommendations; however, it does aim to show readers the different studies on the major topics of discussion surrounding ALDN in order to provide information, and criticize the current management by surgeons. Due to the disparate results in some areas, such as those related to sealants, a critical opinion on the matter must continue to be maintained.
We can conclude that outpatient surgery has proven feasible and, therefore, thanks to appropriate protocols, patient education and close follow-up systems, can be applied with clear benefits.
Regarding the omission of drains, it has been shown that their placement by the surgeon is still a routine common practice. There are multiple disparate articles on the subject with heterogeneous data collected. The technological advances in sealing achieve acceptable rates of symptomatic seroma without drainage. The no drain replacement could be recommended, as it improves autonomy for the patient and reduces the rate of emergency department return visits due to drain malfunction or infection.
The use of sealants remains controversial in a clinical and relevant way. Fibrin sealants appear to achieve greater volume reduction; however, the clinical importance is the percentage of seroma that is clinically relevant and is this the one that should be taken into greater consideration (symptomatic seroma). Due to the variability and heterogeneity of studies (with or without drainage, without sealant, mastectomy, etc.), it is difficult to reach conclusions regarding their use, and specifically which type of sealant. If we have demonstrated that drainage provides no benefits, studies of sealants in patients without drainage are probably necessary, as is not frequent in the literature.
Minimally invasive surgery has demonstrated benefits in clinical practice, in ALDN is not different. We recommend that surgeon learn and train in minimally invasive surgery.
Furthermore, ongoing prospective studies are necessary to establish stronger recommendations.
Comment 5: When discussing different surgical methods and postoperative care strategies, the paper does not fully consider their impact on patients' long-term prognosis.
Response 5: Long term prognosis in ALND is related to the tumor stage, type of tumor adjuvant therapies and other risk factors.
Although several studies have analyzed the effect of drains and tissue sealants on postoperative short term morbidity, none of the available randomized or prospective trials evaluated oncologic outcomes such as local recurrence or long-term survival. Therefore, current evidence does not suggest any impact of these interventions on cancer prognosis affecting in a long term; however, this remains unproven due to the absence of specific data.
We add the corresponding information in each section about long-term prognosis.
Comment 6: The paper's chapter divisions may not be clear enough, with some content overlap between sections. There may be issues with logical coherence when presenting viewpoints and analyzing data.
Response 6:
In order to better structure the manuscript and also to address comment number 2, the sections have been restructured, and a section on clinical considerations has been added (see response 4). Overlapping information between some sections is completely unavoidable due to the heterogeneity of some studies, where we can see the inclusion of patients with drainage and sealant, while in others, the patients do not have drainage but do use sealant. This makes it much more difficult to synthesize and study the factors independently, achieving uniform results.
An attempt has been made to collect the most consistent studies comparing similar cohorts; however, most of the studies mix patients and variables. We believe that studies that consider the use of both sealants and non-drains (which are not frequent) are especially necessary. This would facilitate more independent conclusions regarding the use of sealants or drains alone.
It has been clarified throughout the manuscript, the difficulty of drawing robust conclusions regarding some of the procedures (drains and sealants) due to the heterogeneity of the studies themselves.
Once again, we sincerely appreciate the reviewer’s comments, which have allowed us to significantly improve the quality and clarity of the manuscript.
